

# Deciphering the *Heteropterys pannosa* species complex (Malpighiaceae)

André M. Amorim[1,2], Lucas C. Marinho[3] and Augusto Francener[4]

[1] Departamento de Ciências Biológicas, Universidade Estadual de Santa Cruz, Ilhéus, Bahia, Brazil
[2] Herbário CEPEC, Centro de Pesquisas do Cacau, Ilhéus, Bahia, Brazil
[3] Departamento de Biologia, Universidade Federal do Maranhão, São Luís, Maranhão, Brazil
[4] Departamento de Enfermagem, Faculdade de Goiana, Goiana, Pernambuco, Brazil

## ABSTRACT

We describe three new species of Malpighiaceae that are endemic to central Brazil and related to the *Heteropterys pannosa* complex, a group of xylopodiferous, unbranched subshrubs with fruit in mericarps that have a strongly reduced or no dorsal wing. *Heteropterys tocantinensis* is more common in eastern Tocantins State and on the border with Bahia State, and there are a few records from Mato Grosso State. *Heteropterys veadeirensis* is restricted to northern Goiás State and *H. walteri* has a wider distribution, occurring in some municipalities in northern Goiás and southern Tocantins. Additionally, we also provide detailed redescriptions of *H. pannosa* and *H. rosmarinifolia*, the two previously known species in this complex. All species are considered Endangered (EN) based on IUCN criteria, especially due to the low area of occupancy. Illustrations, distribution maps, and information about phenology and habitat are also provided for all taxa.

## INTRODUCTION

*Heteropterys* Kunth is a highly diverse genus of Malpighiaceae due to its remarkable vegetative and reproductive morphological variability. It comprises approximately 158 species that are distributed from Mexico to South America, including the West Indies (*Anderson, 2001a*; *Anderson, 2013*; *Amorim & Marinho, 2020*). Only *Heteropterys leona* (Cav.) Exell has a disjunct intercontinental distribution. It occurs from Belize to northern Brazil and on the west coast of Africa (*Anderson, 2001a*). The species of *Heteropterys* are subshrubs, shrubs, small trees or robust lianas and has a single synapomorphy: the fruit is a schizocarp where each mericarp or nut has a dominant dorsal wing with a thick inferior margin and lateral wings that are absent or rarely strongly reduced (*Amorim, 2003*; *Pessoa & Amorim, 2016*). The genus was recovered as monophyletic and well-placed in the tetrapteroid lineage, which basically comprises plants with a mericarp with dominant lateral wings and a reduced or no dorsal wing (*Davis & Anderson, 2010*).

Despite the extensive research conducted to solve the problematic taxonomy of *Heteropterys*, in the last 10 years, novelties in this genus have been constantly reported from different habitats. These novelties can often be placed in informal groups supported by lineages in a recent *Heteropterys* phylogeny (C.C. Davis & A.M. Amorim, 2021,

Corresponding author
André M. Amorim,
amorim.uesc@gmail.com

personal communication), which sometimes correspond to infrageneric categories, such as those proposed by *Niedenzu (1903, 1928)*. Notable examples are new taxa of *Heteropterys* recently revealed in the Aptychia (*Amorim & Marinho, 2020*), Metallophyllis (*Amorim et al., 2017*), Parabanisteria (*Almeida & Pellegrini, 2021*), Rhodopetalis (*Pessoa & Amorim, 2016*), Stenophyllarion (*Sebastiani & Mamede, 2010*) and Xanthopetalis groups (*Anderson, 2014*; *Pessoa, Marinho & Amorim, 2019*). Likewise, the resolution of Malpighiaceae species complexes in different genera have been constantly investigated with the objective of clarifying the vegetative and reproductive variability among close species. For example, the resolution of the *Amorimia rigida* complex (*Almeida, Berg & Amorim, 2016*) where three new species were proposed, the *Galphimia langlassei* complex (*Anderson, 2003*) where two new species were proposed, and the *Mascagnia cordifolia* complex (*Anderson, 2005*) and *Mascagnia sepium* complex (*Anderson, 2001b*) where three and seven new species were proposed, respectively. In *Heteropterys* we can highlight the resolution of *Heteropterys anomala* complex (*Amorim, 2003*), where four new species were proposed and *Heteropterys oblongifolia* complex (*Anderson, 1981*) where one new species was proposed but is argued that other taxonomic novelties might arise when a larger number of collections is available to the complex.

The Parabanisteria group is the most diversified lineage of *Heteropterys*. It comprises at least 44 species and occurs in several habitats, such as upland forests, white-sand vegetation and floodplains forests in the Amazon basin. It is also occasionally found in *tabuleiro* and *mussununga* forests (*i.e.*, two kinds of white-sand forest in northeastern Brazil) and a few species occur in inselberg vegetation in the Atlantic Forest domain. However, most species of the Parabanisteria group occur in central Brazil and grow in clay or sand in highland savannas, gallery forests, dry forests and on rock outcrops in the Cerrado domain. Species of this group are recognized by their inflorescence rachis, peduncle, pedicels and sepals covered by a ferrugineous indumentum, eglandular bracteoles at the apex of the peduncle, sepals concealing the petals in bud and revolute at anthesis, and petals spreading and all vivid yellow (*Anderson, 1981*; *Anderson, 2001a*). The mericarp is strongly variable in form and size but generally has a large dorsal wing and lacks lateral wings or crests.

Many species in the Parabanisteria group have morphological characteristics adapted to open vegetation in the Cerrado domain, such as an arborescent or shrubby habit with woody stems, coriaceous leaves and inflorescences that often develop after deciduous leaves period. Among them, *Heteropterys pannosa* Griseb. stands out. It is a subshrub with a xylopodiferous underground stem system, erect and elongate pseudoraceme inflorescences and mericarps without a dorsal wing or with this structure strongly reduced to an apical crest (Figs.1A, 1K and 2A, 2B). *Heteropterys pannosa* was described from a collection made by Johann Emanuel Pohl in Goiás State (*Grisebach, 1858*). Since then, that name has been applied to many collections from different localities at central Brazil (Fig. 3A). Our study clarified the geographical and morphological patterns along the distribution of the *H. pannosa* species complex and that this complex comprises more than just the two species currently recognized: *H. pannosa* and the recently described *H. rosmarinifolia* R.F.Almeida & M.Pell. Thus, in this work we describe three new species

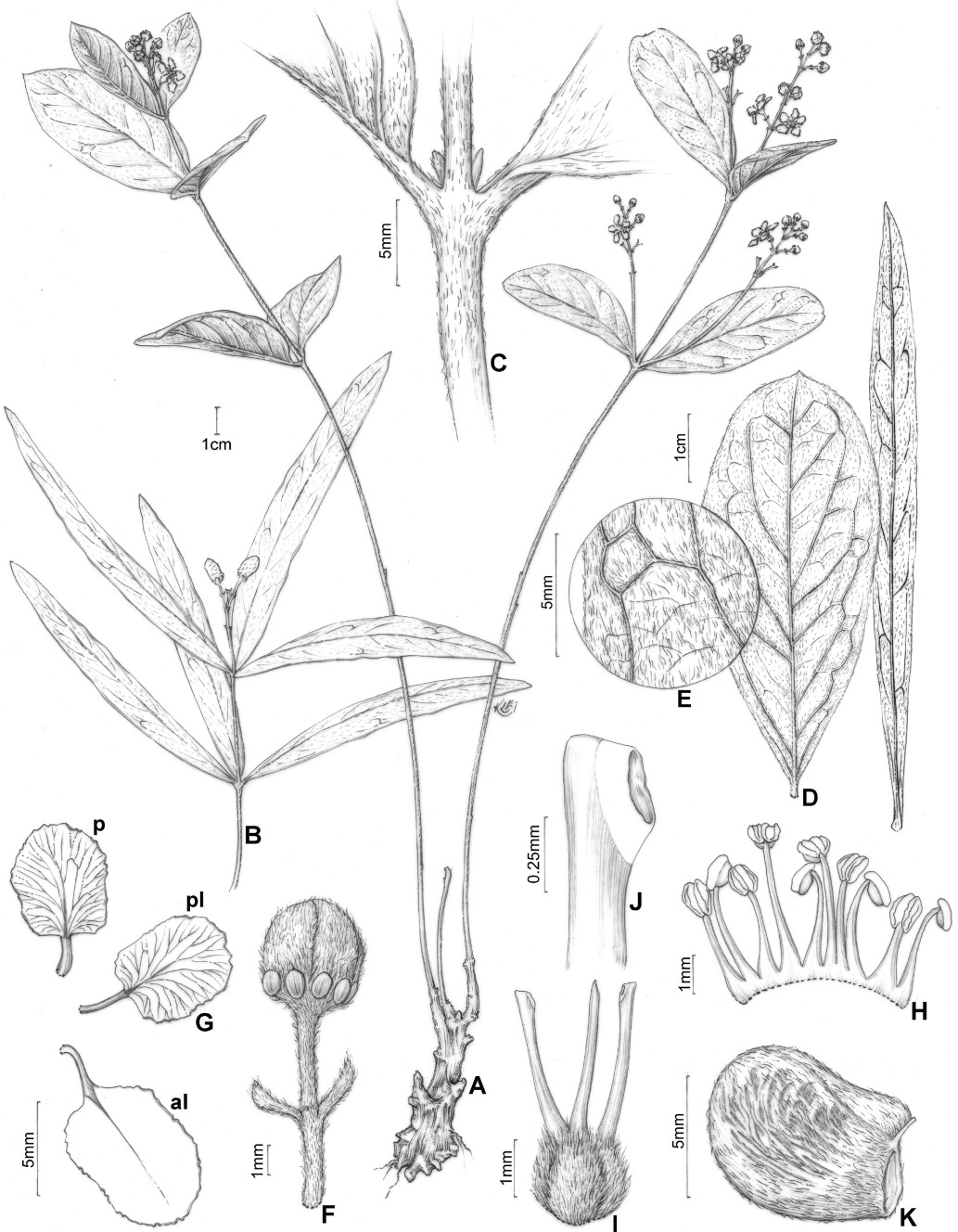

**Figure 1** *H. pannosa. Heteropterys pannosa.* (A) Habit with a xylopodium. (B) Fruiting branch from different individual. (C) Detail of stem and leaf base. (D) Leaves in abaxial view showing the variation in lamina shape. (E) Detail of leaf margin showing the indumentum. (F) Floral bud in lateral view. (G) Petals: posterior–p and posterior-lateral–pl in adaxial view; anterior-lateral–al in abaxial view. (H) Androecium in adaxial view, the stamen second from right opposite the posterior petal, the stamen fourth from left opposite the anterior sepal. (I) Gynoecium showing anterior style at center. (J) Detail of stigma. (K) Mericarp in lateral view. (A, C, E–J from Hatschbach 54683, B from Souza 24795, D from Hatschbach 5468 (left) and Queiroz 15056 (right), K from Moretto 53, by Klei Sousa).

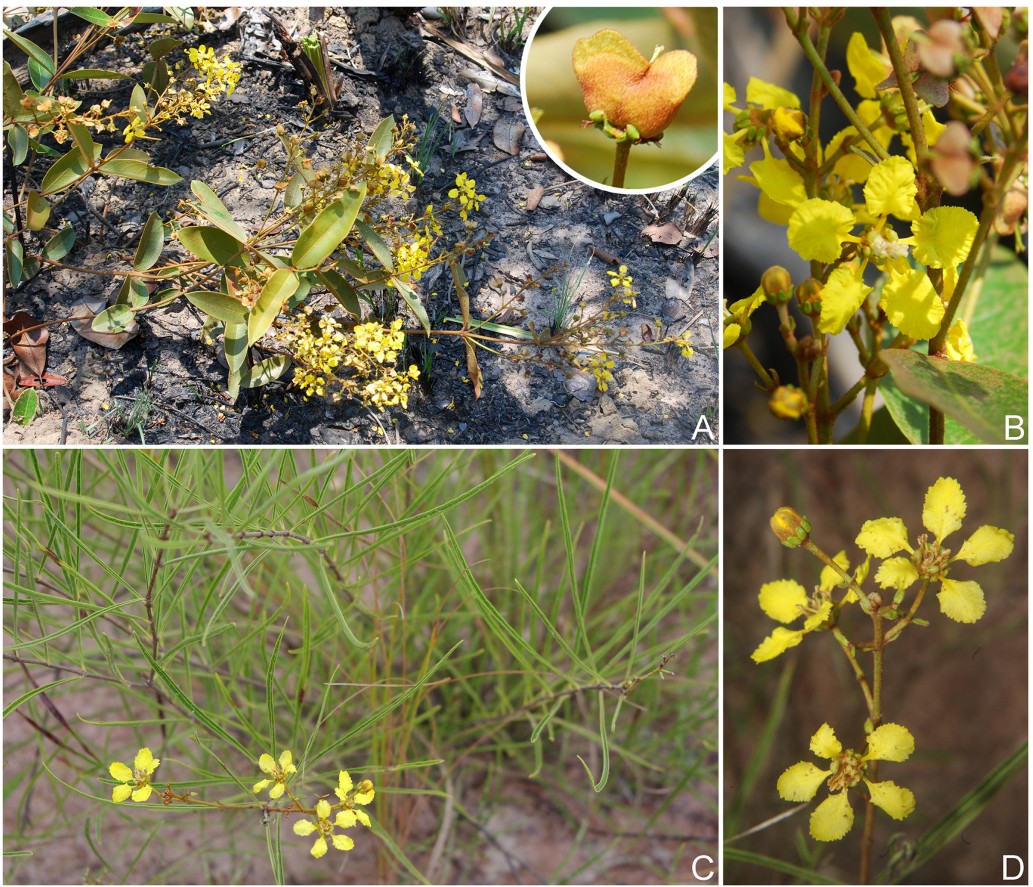

**Figure 2** *H. pannosa* & *H. rosmarinifolia*. *Heteropterys pannosa*. (A) Habit with detail of a nut. (B) Detail of the inflorescence. *Heteropterys rosmarinifolia*. (C) Habit. (D) Detail of an inflorescence. (Photos A–B by HG Silva, C–D by C Proença).

of *Heteropterys* and provide detailed redescriptions of *H. pannosa* and *H. rosmarinifolia*. We also include an identification key, distribution maps, illustrations, and the estimated conservation status for all the species related in the *H. pannosa* complex.

## MATERIALS AND METHODS

We analyzed 82 specimens of the *Heteropterys pannosa* complex from the ALCB, CEPEC, CEN, ESA, HUEFS, MBM, MICH, NY, RB, UB, UESC, UFG, SP and SPF herbaria (acronyms following *Thiers, 2021*-continuously updated). We also used web-based resources, such as the Reflora Virtual Herbarium (Available at http://reflora.jbrj.gov.br and accessed in November 2021) and SpeciesLink (Available at https://specieslink.net/ and accessed in November 2021), to check additional specimens (as digital images), including types. Herbaria consulted by digital images are indicated by an asterisk in specimens examined. Descriptions of the characters are based on dried material. The geographic distributions maps were created using the website SimpleMappr (*Shorthouse, 2010*), with subsequent style modifications. The conservation status of each species was assessed using *IUCN Standards & Petitions Subcommittee (2017)* guidelines and criteria; the area of occupancy and extent of occurrence were calculated using GeoCAT (*Bachman et al., 2011*).

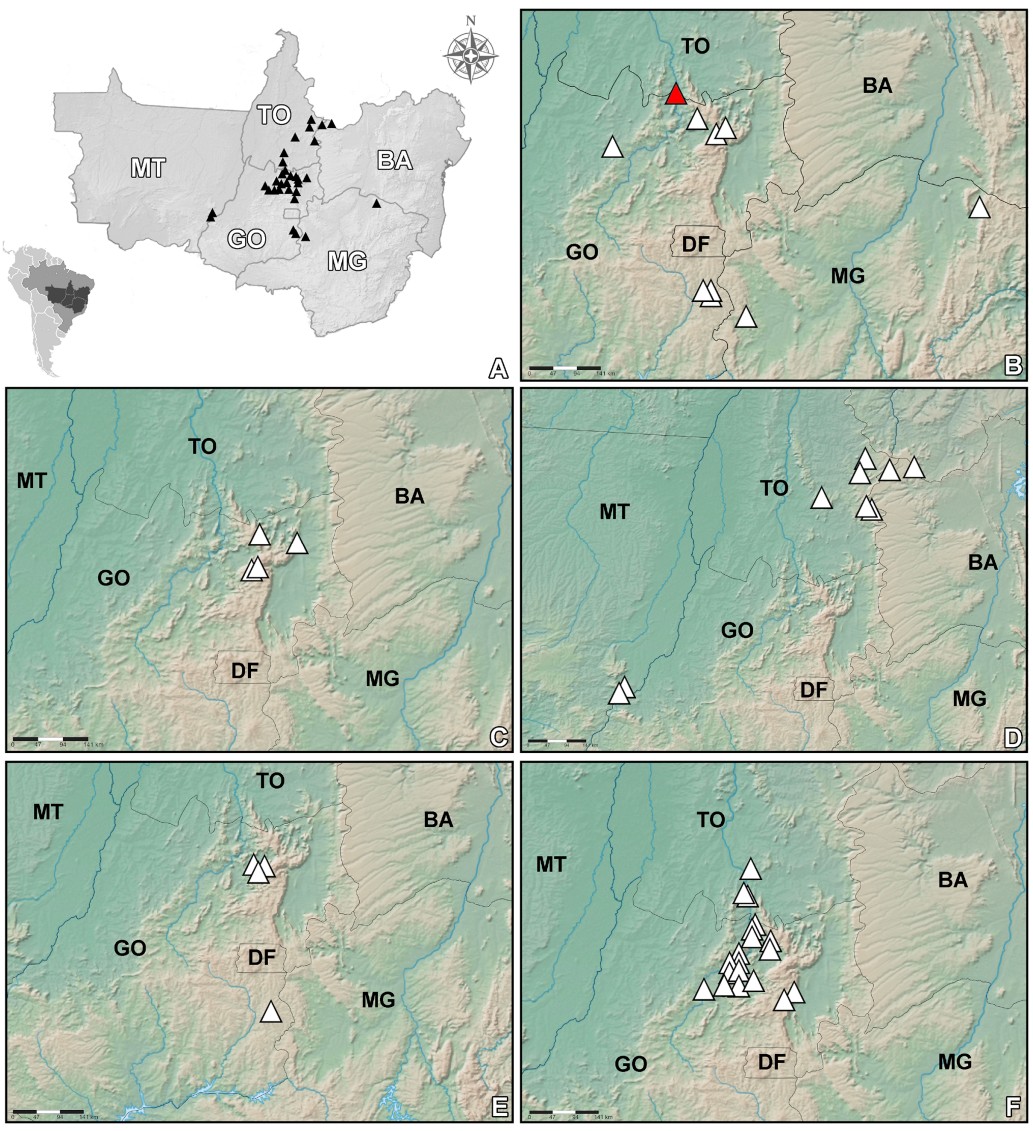

**Figure 3 Map *Heteropterys pannosa* complex.** Geographic distribution of *Heteropterys pannosa* and related species. (A) General distribution of the *H. pannosa* complex by Brazilian states based in all specimens. (B) Distribution of the *H. pannosa* (red triangle shows the probable type locality of this species). (C) Distribution of the *H. rosmarinifolia*. (D) Distribution of the *H. tocantinensis*. (E) Distribution of the *H. veadeirensis*. (F) Distribution of the *H. walteri*.

The electronic version of this article in Portable Document Format (PDF) will represent a published work according to the International Code of Nomenclature for algae, fungi, and plants (ICN), and hence the new names contained in the electronic version are effectively published under that Code from the electronic edition alone. In addition, new names contained in this work which have been issued with identifiers by IPNI will eventually be made available to the Global Names Index. The IPNI LSIDs can be resolved and the associated information viewed through any standard web browser by appending the LSID contained in this publication to the prefix "http://ipni.org/".

The online version of this work is archived and available from the following digital repositories: PeerJ, PubMed Central SCIE, and CLOCKSS.

## RESULTS

### Key to the species of the *Heteropterys pannosa* complex

1. Young stems, petiole and lamina glabrous . . . . . . . . . . . . . . . . . . . . . . . . . . . . . . . . . . *H. veadeirensis*

1.′ Young stems, petiole and lamina covered by indumentum with simple and basifixed or Y-V-T-shaped trichomes . . . . . . . . . . . . . . . . . . . . . . . . . . . . . . . . . . . . . . . . . . . . . . . . . . . . . . . . 2

2. Young petiole, midrib of lamina, inflorescence rachis, peduncle and pedicel covered by hispid indumentum . . . . . . . . . . . . . . . . . . . . . . . . . . . . . . . . . . . . . . . . . . . . . . . . . . . *H. walteri*

2.′ Young petiole, midrib of lamina, inflorescence rachis, peduncle and pedicel covered by sericeous or tomentose indumentum. . . . . . . . . . . . . . . . . . . . . . . . . . . . . . . . . . . . . . . . . 3

3. Lamina of larger leaves linear or narrowly oblong-oblanceolate, strongly conduplicate and arranged on stem with short internodes (< 2 rarely 2.5 cm long) . . . . . . . . . . . . . . . . . . . . . . . . . . . . . . . . . . . . . . . . . . . . . . . . . . . . . . . . *H. rosmarinifolia*

3.′ Lamina of larger leaves elliptical, lanceolate, oblong, obovate to ovate, not conduplicate and arranged on stem with large internodes (≥ 2 cm long) . . . . . . . . . . . . . . . . . . . . . . . . 4

4. Peduncle shorter than pedicel at anthesis, anthers irregularly pilose, styles dorsally apiculate at apex . . . . . . . . . . . . . . . . . . . . . . . . . . . . . . . . . . . . . . . . . . . . . . . . . . . *H. tocantinensis*

4.′ Peduncle generally equaling pedicels or longer at anthesis, anthers glabrous, styles dorsally obtuse or truncate at apex. . . . . . . . . . . . . . . . . . . . . . . . . . . . . . . . . . . . . . . . *H. pannosa*

*Heteropterys pannosa* Griseb. in Mart. Fl. Bras. 12(1): 70. 1858.
(Figs. 1 and 2A, 2B)

### *Type*

[Brazil], Goyaz: Serra S. Felis bei Rio Custodio, *J.E. Pohl* 1940 [d 1 5 3 0] (Holotype: BR barcode BR000000986917!, Isotypes: K barcode K000427071!, W-0069487!, W-0069488!).

### *Description*

Subshrub, 0.2–0.8 m tall, stems erect, cylindrical, 1–2(–3) mm diam., sericeous to glabrate with age, lenticels not seen, unbranched or nearly so, all arising from a xylopodium. Leaves opposite or occasionally 3-whorled on the same stem, internodes 2–4(–6.8) cm long; petiole (1.5–)3–5 mm long, densely sericeous to glabrate with age, eglandular; stipules not seen; lamina of larger leaves 5.7–9(–11.7) cm long, (0.7–)1.1–3 cm wide, densely sericeous or tomentose-ferrugineous to irregularly glabrate with age on basal leaves and densely and persistently sericeous or tomentose-ferrugineous on leaves closer to the inflorescence, subcoriaceous to coriaceous, not conduplicate, elliptical, lanceolate or oblong to slightly obovate, the base cuneate, rarely obtuse or attenuate, the apex obtuse or sometimes (on the same stem) gradually tapered and becoming acute or acuminate, the margins entire or rarely slightly revolute, glands in abaxial surface absent or inconspicuous, hidden by indumentum, the lateral veins and reticulum prominent, especially on the abaxial surface.

Inflorescence a pseudoraceme, axillary or terminal, erect, 3–6.6 cm long, densely sericeous-ferrugineous, sometimes glabrate with age, with irregular internodes between each 1–2 flowers, mostly comprising (4–)10–16 flowers distributed throughout the rachis; bracts 3–4 mm long, 1–1.5 mm wide, densely sericeous, lanceolate, margins entire, eglandular, persistent; peduncle 4–7(–10) mm long, 0.5–1 mm wide, densely sericeous-ferrugineous; bracteoles apical, 1.5–2 mm long, 0.8–1.3 mm wide, ovate-lanceolate, persistent, eglandular; pedicel 4–6(–10) mm long, 0.5–1 mm wide, uniformly slender, densely sericeous-ferrugineous. Sepals 4.5–5.5 mm long, 1.5–2.5 mm wide, ovate, acute at apex, revolute at anthesis, not appressed against filaments at anthesis, abaxially sericeous-ferrugineous, adaxially glabrous, the anterior sepal eglandular, the four lateral sepals biglandular, the glands 1–2 mm diam., green. Petals not exposed in the enlarging bud, vivid yellow, glabrous, membranaceous, not keeled, irregularly erose and eglandular at the margin, the posterior-lateral and anterior-lateral petals similar to each other, spreading, the claw 1.5–3 mm long, the limb 3–5 mm long, 2.5–5 mm wide; posterior petal spreading, the claw 1–2 mm long, the limb 3.4–5 mm long, 2.4–4.5 mm wide. Stamens with filaments strongly heteromorphic, longer opposite sepals than opposite petals, glabrous, 2.6–4.4 mm long, 0.9–1 mm wide at base, all straight and slender, basally connate; anthers 1.1–1.3 mm long, glabrous, slightly reflexed at anthesis, all alike; the connective uniformly brown. Ovary 1–1.4 mm tall, densely sericeous-ferrugineous; styles 3–3.7 mm long, slightly unequal, larger than the largest stamens, the anterior style straight and the two posterior styles slightly divergent at base, glabrous, dorsally obtuse or truncate at apex; stigmas lateral, all three facing the center of the flower. Mericarp with ellipsoidal nut, 7–11 mm long, 5–7 mm wide, with parallel longitudinal veins on each side, sericeous to glabrate with age; lateral wings or crests absent; dorsal wing absent or strongly reduced and not arising at style, ca. 1 mm wide; ventral areole 3–4 mm tall, ca. 2.5 mm wide, ovate.

### Distribution, phenology and conservation status

*Heteropterys pannosa* occurs in Goiás State and for the first time is recorded for Minas Gerais State (Fig. 3B). It grows in sandy soils (*i.e.*, quartzite and plinthite formation, Fig. 2A) in grassy meadows with subshrubs and small shrubs, and on rocky outcrops on the highest and driest slopes, between 400 and 1.700 m a.s.l. *Heteropterys pannosa* is often found in vegetation associated with *Manihot* spp. (Euphorbiaceae). This species has been collected with buds and/or flowers in March, April, June and from August to December. Fruits have been collected from November to April. Although it has a wide extant of occurrence, *H. pannosa* is Endangered (EN) due to an area of occupancy less than 40 km$^2$ [B2aii].

### Additional Specimens Examined

Brazil. Goiás: Mun. Alto Horizonte, região da Sururuca, Fazenda Cajás, 14°11′49″S, 49°16′42″W, ca. 404 m, 22 August 2016 (bud, fl), *J.E.Q. Faria* 6470 (HDJF*, UB); Mun. Alto Paraíso de Goiás, Chapada dos Veadeiros, ca. 25 km by road N of Alto Paraíso, 13°53′59.1″S, 47°23′48.9″W, ca. 1.700 m, 8 March 1973 (ste), *W.R. Anderson et al.* 6673 (MICH, NY, UB); ca. 40 km by road of Alto Paraíso, 13°53′59.1″S, 47°23′48.9″W, ca. 1.500 m,

10 March 1973 (fr), *W.R. Anderson et al.* 6769 (MICH, NY, UB); Mun. Cavalcante, RPPN Serra do Tombador, 13°40′01″S, 47°48′04″W, without date (fl), *C.B.R. Munhoz et al.* 7971 (UB); RPPN Serra do Tombador, 13°38′04″S, 47°49′06″W, ca. 857 m, 29 October 2011 (fl), *A.R.O. Ribeiro et al.* 278 (UB); Mun. Cristalina, 10 km by road N of Cristalina, 13°53′59.1″S, 47°23′48.9″W, ca. 1,080 m, 3 April 1973 (fl, fr), *W.R. Anderson et al.* 8049 (MICH, NY, UB); ca. 5 km em direção a Brasília, margem da Rodovia BR-040, 16°45′S, 47°40′W, 29 July 2007 (bud), *M.A. Silva* 6155 (IBGE*, UB); ca. 5 km em direção a Brasília, lado esquerdo da BR-040, 7 January 2008 (fr), *M.A. Silva* 6314 (CEPEC, IBGE*); ca. 5 km of Cristalina, 17°S, 48°W, ca. 1.175 m, 2 November 1965 (fr), *H.S. Irwin et al.* 9794 (NY, RB, UB); Serra dos Cristais, ca. 3 km S of Cristalina, 17°S, 48°W, ca. 1.200 m, 3 March 1966 (fl), *H.S. Irwin et al.* 13381 (NY, UB); Serra dos Cristais, 5 km by road E of Cristalina, ca. 1.200 m, 5 April 1973 (ste), *W.R. Anderson* 8168, 8169, 8170, 8171 (MICH, UB); Serra dos Cristais, 6 km de Cristalina em direção a Unaí, GO-309, 10 September 1998 (bud, fl), *V.C. Souza et al.* 21412 (CEN, CEPEC, ESA, RB, SP); Serra dos Cristais, 6 km de Cristalina em direção a Unaí, GO-309, 10 September 1998 (old fl), *V. C. Souza et al.* 21378 (CEPEC, ESA, RB); Serra dos Cristais, 5 km S of Cristalina, 17°S, 48°W, ca. 1.175 m, 1 November 1965 (bud, fl), *H.S. Irwin et al.* 9735 (NY, UB); RPPN Linda Serra dos Topázios, 16°45′S, 47°40′W, 29 October 1995 (bud, fl), *C. Proença & G. L. Moretto* 1315 (CEPEC, UB); Linda Serra dos Topázios, 16°45′S, 47°40′W, 26 October 1996 (ste), *C. Proença & R.S. Oliveira* 1564 (UB); 16°45′S, 47°40′W, ca. 461 m, 9 km by road S of Catalão, 4 April 1973 (old fl), *W.R. Anderson* 8092 (MICH, UB); Serra dos Cristais, RPPN Linda Serra dos Cristais, trilha que leva ao Poço da Diretoria, 16°45′S, 47°40′W, 23 March 1996 (fr), *G. L. Moretto* 53 (UB); estrada de terra a NE de Cristalina, ca. 1.6 km da cidade, 16°44′49″S, 47°37′28″W, ca. 1.170 m, 16 December 2014 (fl), *J.B.A. Bringel et al.* 1137 (CEN); Mun. Teresina de Goiás, descida para Cavalcante, 17 October 1990 (bud, fl), *G.M. Hatschbach & J. M. Silva* 54683 (CEPEC, MBM, NY); GO-309, ca. 5 km do Mun. Teresina de Goiás, ca. 40 km N Alto Paraíso de Goiás, a NE de Cristalina, 16°42′37″S, 47°35′23″W, 15 December 2014 (fr), *J.B.A. Bringel et al.* 1127 (CEN, CEPEC); 13°53′59.1″S, 47°23′48.9″W, ca. 1.500 m, 16 March 1973 (bud, fl, fr), *W.R. Anderson et al.* 7156 (MICH, NY, UB), Fazenda Hotel Ecológico Alpes Goianos, GO-118, Km 202, 13°53′59.1″S, 47°23′48.9″W, 31 July 2000 (fr), *V.C. Souza et al.* 24795 (CEPEC, ESA, RB). Minas Gerais. Mun. Monte Azul, Serra da Formosa em frente ao Pico da Formosa, 15°13′48″S, 42°48′14″W, 27 October 2010 (bud, fl), *L.P. Queiroz et al.* 15036 (CEPEC, HUEFS); Mun. Paracatu, ramal entrando a NE da BR-040, 16°47′58″S, 47°34′00″W, 30 October 2010 (bud, fl), *L.P. Queiroz et al.* 15056 (HUEFS, RB).

### Remarks

The type collection of *Heteropterys pannosa*, gathered by Johann Emanuel Pohl (1782–1834) in Goiás State, does not have a date and is annotated with the enigmatic locality of "Serra S. Felis bei Rio Custódio." Based on the travel diary of Johann Emanuel Pohl that narrates the details of his journey in Brazil (*Pohl, 1976*), we assume that the type was collected on 8 July 1819, the only time Pohl was in this locality. In this diary, Pohl also notes that the Custódio River is near the Traíras and Maranhão rivers (*Pohl, 1976*).

An antique map of Goiás State (*ArPDF, 2020*) shows "Chapada de São Félix" (*i.e.*, annotated by Pohl as "Serra de S. Felis"). This plateau is near the Maranhão River, which is actually denominated the Tocantins River. The point on the map for Goiás State (*ArPDF, 2020*) is at 13° latitude. We carefully looked at this region using Google Maps® and found the Custódio River, which is part of the Tocantins River hydrographic basin. Thus, this region is probably the *H. pannosa* type collection locality and is delimited by the municipalities of Minaçu in Goiás State and Paranã in Tocantins State.

The vegetative morphology of *Heteropterys pannosa* is extremally variable (Figs. 1A–1D). In the type (*Pohl* 1940) and several other collections (*e.g.*, *Souza* 24795, Fig. 1B), the shape of the leaf lamina is elliptic. In other collections, the lamina is lanceolate (*e.g.*, *Queiroz* 15056, Fig. 1D) or oblong-obovate (*e.g.*, *Hatschbach* 54683, Figs. 1A, 1D). Further, many collections have different lamina shapes on the same stem (*e.g.*, especially *Anderson* 6673) and a large variation in the indumentum density caused by the gradual loss of trichomes. For these reasons, the absence of diagnostic differences in reproductive characteristics and the fact that many individuals occur in sympatry in some areas (C. Anderson, 2003, pers. comm.), we decided to define *H. pannosa* within a broad concept. This position should be re-evaluated when extensive fieldwork can be conducted throughout the area of occurrence and there are more and better specimens are available for study. The vegetative and reproductive characters of *H. pannosa* are illustrated for the first time.

***Heteropterys rosmarinifolia*** R.F.Almeida & M.Pell. PhytoKeys 175: 47. 2021. (Figs. 2C–2D and 4)

*Type*
Brazil. Goiás: Mun. Cavalcante, Reserva Natural Serra do Tombador, road GO-241, estrada de terra para o Engenho II, a direita da estrada, 13°42′S, 47°48′W, 25 July 2014 (fl), *R. Sartin et al.* 576 (Holotype: UFRN barcode UFRN00024927!; Isotype: RB barcode RB01408371!).

*Description*
Subshrub, 0.2–0.8(–1.2) m tall, stems erect, cylindrical, 1.5–2 mm diam., densely sericeous to glabrate with age, developing scattered lenticels, unbranched or nearly so, all arising from a xylopodium. Leaves opposite or usually 3–4-whorled on the same stem, internodes 0.3–2(–2.5) cm; petiole 1–3 mm long, sericeous to glabrate, eglandular; stipules ca. 0.5 mm long, persistent, generally hidden by indumentum; lamina of larger leaves 1.3–11.4 cm long, 0.1–1.3 cm wide, abaxial surface very sparsely sericeous to glabrate with age or densely and persistently sericeous on leaves near the inflorescence, adaxial surface glabrate, subcoriaceous to coriaceous, strongly conduplicate, linear or narrowly oblong-oblanceolate, the base acute, the apex acuminate or rarely acute, the margins entire to slightly revolute, abaxial surface sometimes with two large glands at base and usually a row of smaller impressed and inconspicuous glands near or somewhat inside the margin, glands rarely absent, the lateral veins and reticulum prominent on both surfaces. Inflorescence a pseudoraceme, axillary or terminal, erect, (1–)3.5–10.5 cm long, densely

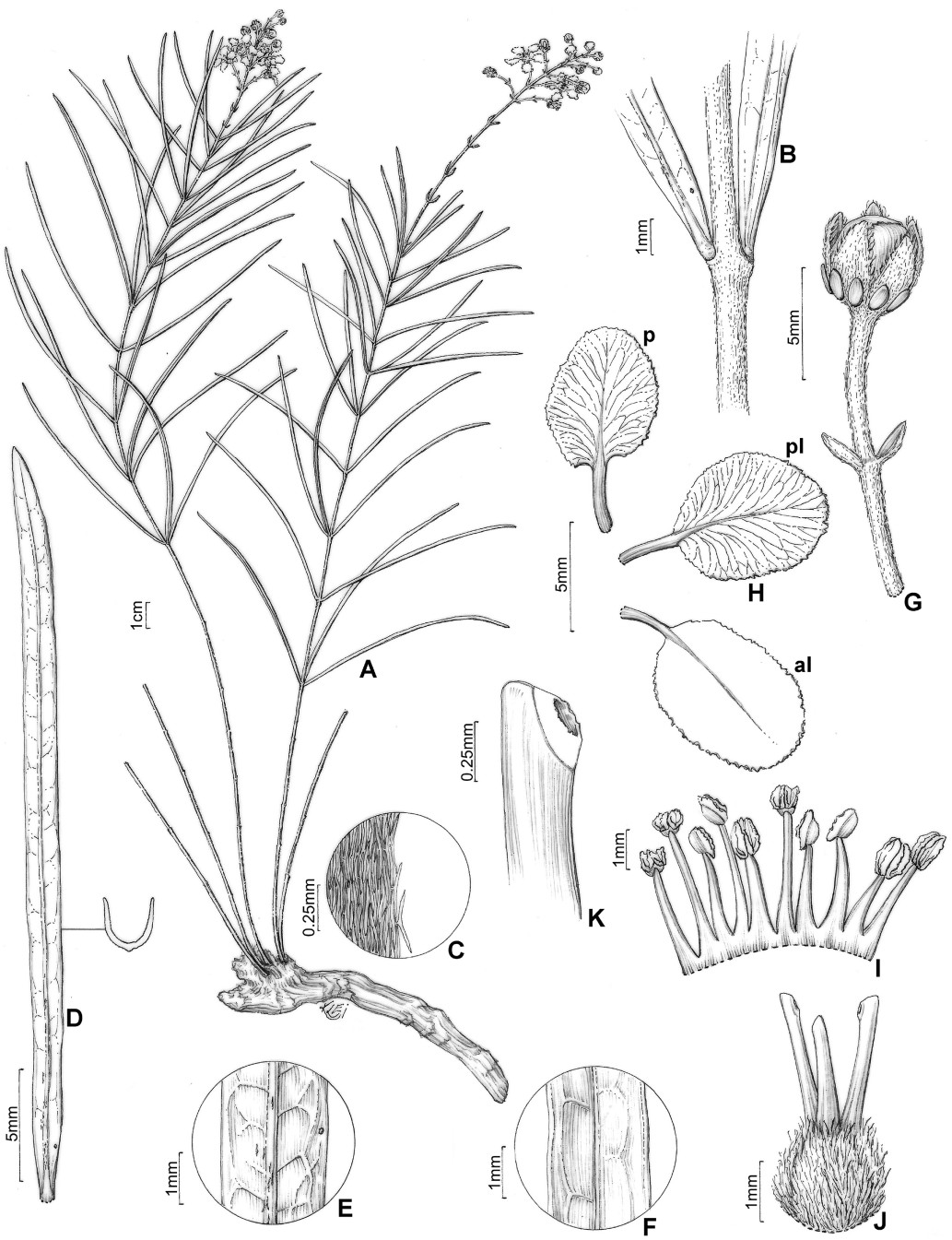

**Figure 4** *H. rosmarinifolia.* *Heteropterys rosmarinifolia.* (A) Habit with a xylopodium. (B) Detail of stem and leaf base. (C) Detail of stem, showing the indumentum. (D) Leaf in abaxial view and detail of a transversal section. (E) Detail of leaf in abaxial view. (F) Detail of leaf in adaxial view. (G) Floral bud in lateral view. (H) Petals: posterior–p and posterior-lateral–pl in adaxial view; anterior-lateral–al in abaxial view. (I) Androecium in adaxial view, the stamen second from right opposite the posterior petal, the stamen fourth from left opposite the anterior sepal. (J) Gynoecium showing anterior style at center. (K) Detail of stigma. (A–K from Pirani 6421, by Klei Sousa).

sericeous-ferrugineous, with irregular internodes between each pair of flowers, mostly comprising (4–)10–16 flowers distributed throughout the rachis; bracts 2–4 mm long, 0.7–1 mm wide, sparsely sericeous, linear-lanceolate, margins entire, eglandular or with 1–2 glands at base, the glands ca. 0.2 mm diam., persistent; peduncle 6–10 mm long, sericeous-ferrugineous; bracteoles apical, 1–2.8 mm long, 0.5–1 mm wide, sparsely sericeous, lanceolate, persistent, eglandular or with 1–2 glands at base, the glands ca. 0.2 mm diam.; pedicel 5–9(–11.5) mm long, 0.5–1.1 mm wide, uniformly slender, sericeous-ferrugineous. Sepals 4–5.5 mm long, 1.5–2.3 mm wide, narrowly ovate, acute at apex, revolute at anthesis, not appressed against filaments at anthesis, abaxially sericeous-ferrugineous, adaxially green and glabrous, all eglandular or all biglandular or the anterior sepal eglandular and the four lateral sepals biglandular, the glands 1–2 mm diam., green. Petals not exposed in the enlarging bud, vivid yellow, glabrous, membranaceous, not keeled, irregularly erose and eglandular at the margin, the posterior-lateral and anterior-lateral petals similar to each other, spreading, the claw 1–3.7 mm long, the limb 3.5–6 mm long, 2.5–5.5 mm wide; posterior petal spreading, the claw 1–2.5 mm long, the limb 3.4–5.3 mm long, 2.4–4 mm wide. Stamens with filaments slightly heteromorphic, longer opposite sepals than opposite petals, glabrous, 2.5–4.5 mm long, 0.9–1 mm wide, all straight and slender, basally connate; anthers 1–1.2 mm long, glabrous, irregularly reflexed at anthesis, all alike; the connective uniformly yellow. Ovary 1.1–1.5 mm tall, sericeous-ferrugineous; styles 3–3.5 mm long, slightly unequal, larger than the largest stamens, the anterior style erect and straight, the two posterior styles slightly divergent, glabrous, obtuse at apex; stigmas lateral, all three facing the center of the flower. Fruit unknown.

### Distribution, phenology and conservation status
The recently described *Heteropterys rosmarinifolia* was originally known from two specimens collected in the Serra do Tombador Natural Reserve (*Almeida & Pellegrini, 2021*) and was therefore assessed as data deficient (DD) by the authors. We increased the distribution area of the species to another protected area, Chapada dos Veadeiros National Park, which is also in northern Goiás State (Fig. 3C). An extent of occurrence less than 3.000 km$^2$ and an area of occupancy less than 16 km$^2$ means *H. rosmarinifolia* is Endangered (EN) based on *IUCN Standards & Petitions Subcommittee (2017)* criteria, even though it occurs in two environmental protection areas [B1 + B2a]. *Heteropterys rosmarinifolia* grows in sandy savannas (Fig. 2C), between 1.100 and 1.300 m a.s.l. This species has been collected with buds and/or flowers from July to September.

### Additional Specimens Examined
Brazil. Goiás: Mun. Alto Paraíso de Goiás, Parque Nacional da Chapada dos Veadeiros, Fazenda São Bento, Córrego Almécegas, 14°09′58″S, 47°35′31″W, 10 August 2007 (bud, fl), *C. Proença & S.A. Harris* 3384 (SP, UB); Fazenda São Bento, entre as Cachoeiras Almécegas I e São Bento, 14°10′22″S, 47°35′27″W, ca. 1.110 m, 30 June 2018 (bud), *P.Q. Rosa et al.* 2257 (HEPH*, UB); Rodovia Alto Paraíso de Goiás a Brasília, GO 118, 4 km do trevo sul de Alto Paraíso de Goiás, 14°10′37″S, 47°30′55″W, ca. 1.280 m, 04 September

2013 (bud, fl), *J.R. Pirani et. al.* 6421 (CEPEC, SPF); Mun. Cavalcante, caminho para a Cachoeira da Ave-Maria, ponto onde se vê a cachoeira, 13°44′26″S, 46°52′46″W, 22 September 2015 (fl), *L. Rocha et al.* 668 (Paratypes: CEPEC, HUEFS).

### Remarks

*Heteropterys rosmarinifolia* is redescribed here due to the analysis of a larger number of collections. As noted by *Almeida & Pellegrini (2021)*, this species is related to the *Parabanisteria* group and very closest to *H. pannosa*, although collections with fruits have not been found. *Heteropterys rosmarinifolia* can be distinguished from the other species of the *H. pannosa* complex by the linear or narrowly oblong-oblanceolate and strongly conduplicate laminae arranged on short internodes. The vegetative and reproductive characters of this species are illustrated for the first time. Some herbarium collections of *H. rosmarinifolia* were misidentified as *Byrsonima linearifolia* A. Juss., a species with linear and strongly conduplicate leaves, also present in the state of Goiás.

### *Heteropterys tocantinensis* Amorim & Francener *sp. nov.*
(Figs. 5 and 6A, 6B)

### Type

Brazil. Tocantins: Mun. Mateiros, Região do Jalapão, estrada Mumbuca a Boa Esperança, próximo ao posto Naturantins, 10°23′38″S, 46°36′46″W, 8 December 2005 (bud, fl), *G.H. Rua et al.* 681 (Holotype: CEN!; Isotype: CEPEC!).

### Diagnosis

*Heteropterys tocantinensis* differs from the other species in the *H. pannosa* complex in its peduncle, which is shorter than the pedicel at anthesis (*vs* peduncle generally equaling the pedicel or rarely longer), anthers irregularly pilose (*vs* glabrous), styles dorsally apiculate (*vs* hooked, obtuse or truncate) and dorsal wing of mericarp present and 5–6 mm wide (*vs* absent or rarely reduced to an apical crest and 1–3 mm wide).

### Description

Subshrub, ca. 1 m tall, stems erect, cylindrical, 2–4 mm diam., densely sericeous to glabrate with age, developing small, scattered lenticels, all arising from a xylopodium. Leaves opposite, arranged in internodes 3–4.6 cm long; petiole 2–3 mm long, sericeous to glabrate with age, eglandular; stipules not seen; lamina of larger leaves (4.8–)6–9(–11.2) cm long, 2.8–6 cm wide, adaxially sparsely sericeous, abaxially densely sericeous to early glabrate, the midrib, primary, and secondary veins on both surfaces sericeous, coriaceous, oblong to slightly obovate, the base obtuse to cordate, the apex obtuse or slightly cuspidate, the margins entire, sometimes with two glands abaxially at base near the midrib and usually with 6–8 smaller impressed glands in an inframarginal row on each side of the lamina, the glands ca. 0.5 mm diam., the lateral veins and reticulum strongly prominent on both surfaces. Inflorescence a pseudoraceme, mostly elongate, axillary or terminal, erect, 5–11 cm long, sericeous to tomentose, with measurable and irregular internodes between groups of 2–3 flowers, mostly comprising 6–21 flowers distributed throughout the rachis; bracts 2–3 mm long, ca. 0.7 mm wide, abaxially tomentose, ovate, margins entire,

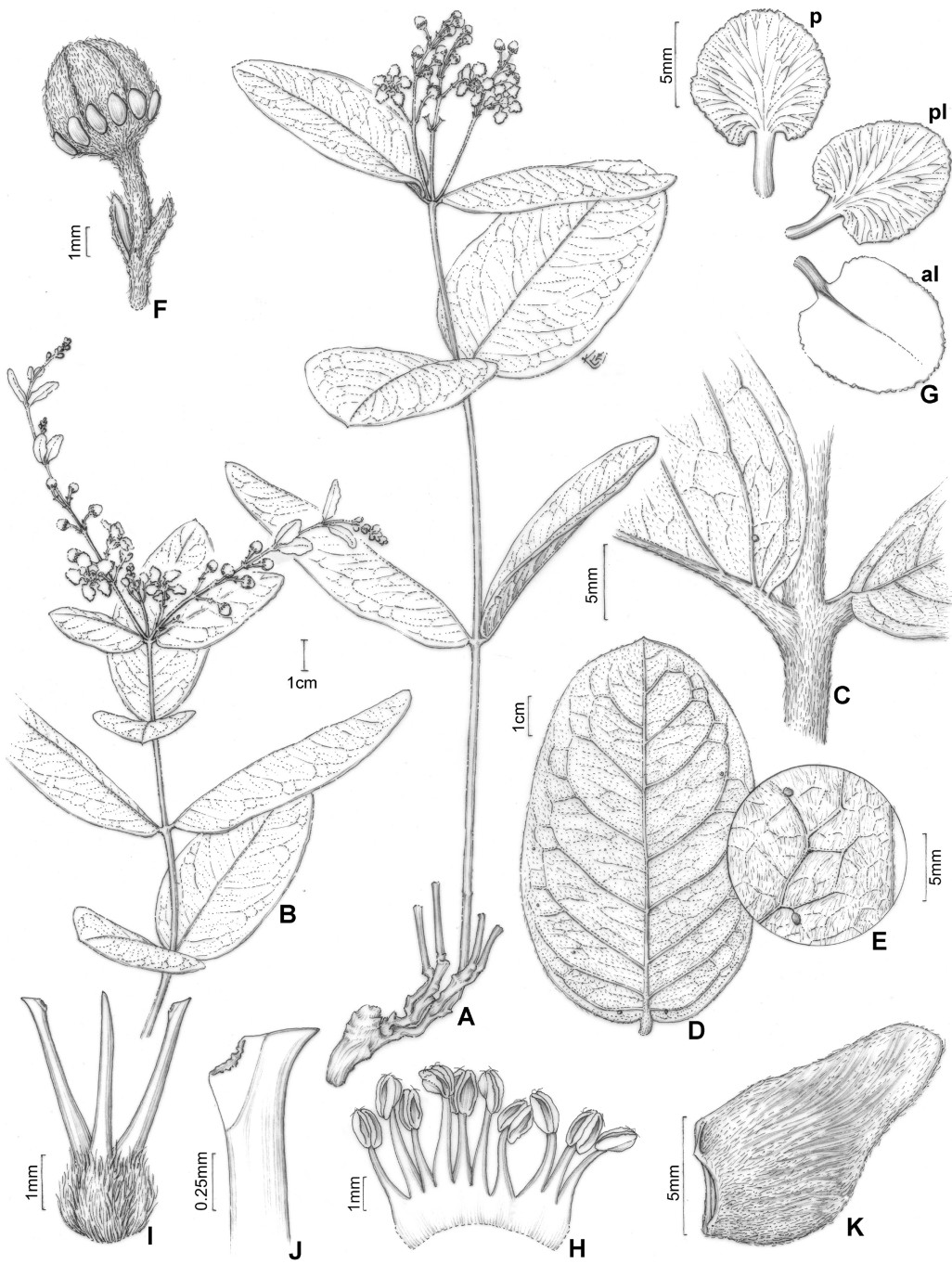

**Figure 5 *H. tocantinensis*.** *Heteropterys tocantinensis*. (A) Habit with a xylopodium. (B) Flowering branch from different individual. (C) Detail of stem and leaf base. (D) Leaf in abaxial view. (E) Details of leaf margin showing the glands, abaxial view. (F) Floral bud in lateral view. (G) Petals: posterior–p and posterior-lateral–pl in adaxial view; anterior-lateral–al in abaxial view. (H) Androecium in adaxial view, the stamen second from right opposite the posterior petal, the stamen fourth from left opposite the anterior sepal. (I) Gynoecium showing anterior style at center. (J) Detail of stigma. (K) Mericarp in lateral view. (A, C–J from Rua 681, B from Amorim 9196, K from Rua 680, by Klei Sousa).

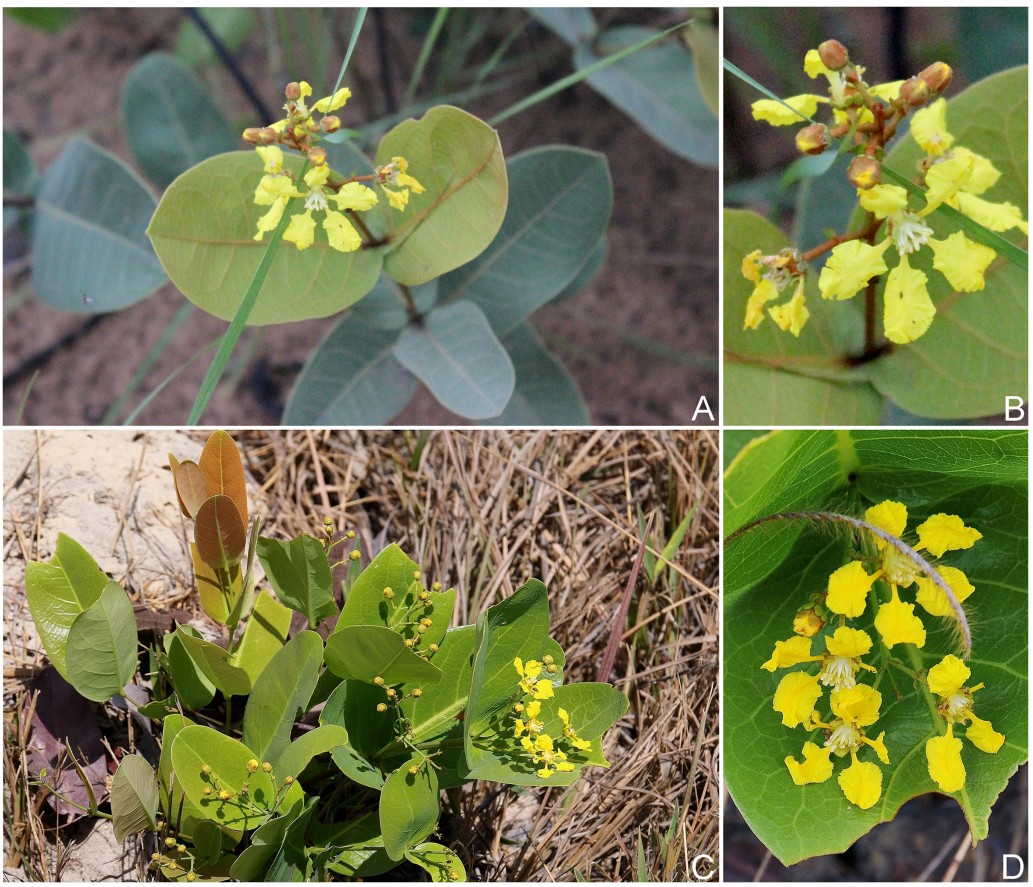

**Figure 6 H. tocantinenses & H. veadeirensis.** *Heteropterys tocantinensis* (A) Habit. (B) Detail of an inflorescence. *Heteropterys veadeirensis* (C) Habit. (D) Detail of an inflorescence. (Photos A–B by GM Antar, C–D by M Mercadante).

eglandular, persistent; peduncle 2–4 mm long, ca. 1 mm wide, tomentose; bracteoles apical, 1.5–2 mm long, ca. 0.8 mm wide, abaxially tomentose, ovate, eglandular, persistent; pedicel 6–8 mm long, ca. 1 mm wide, uniformly slender, tomentose. Sepals 3–3.3 mm long, 1.5–1.7 mm wide, narrowly ovate, acute at apex, strongly revolute at anthesis, not appressed against filaments at anthesis, abaxially tomentose, adaxially glabrous, the anterior sepal eglandular, the four lateral sepals biglandular, the glands 0.7–1.2 mm diam. Petals not exposed in the enlarging bud, vivid yellow, glabrous, membranaceous, not keeled, erose and eglandular at the margin, the posterior-lateral and anterior-lateral petals similar to each other, spreading, the claw 3–3.5 mm long, the limb 7–7.5 mm long, 6–6.5 mm wide; posterior petal suberect, the claw 3–4 mm long, the limb 5.5–6 mm long, 4–5 mm wide. Stamens with filaments slightly heteromorphic, longer opposite sepals than opposite petals, glabrous, 3–4.2 mm long, 0.3–0.5 mm wide, all straight and slender, basally connate; anthers 0.7–1 mm long, irregularly pilose, slightly reflexed at anthesis, all alike; the connective proximally dark brown, distally pale yellow. Ovary 2–2.5 mm tall, densely sericeous-ferrugineous; styles 3.5–4.5 mm long, slightly unequal, larger than the largest stamens, all straight, somewhat divergent, glabrous, dorsally apiculate at apex;

stigmas lateral, all three facing the center of the flower. Mericarp with rounded nut, 6–8 mm long, ca. 6 mm wide, with inconspicuous longitudinal veins, densely sericeous, the trichomes persistent; lateral wings or crests absent; dorsal wing reduced and arising slightly at style, 5–6 mm wide, widest near distal side of nut; ventral areole ca. 3 mm tall, ca. 2.8 mm wide, rounded.

### Distribution, phenology and conservation status

*Heteropterys tocantinensis* occurs in the state of Tocantins, on the border of the states of Bahia and Maranhão. A few records from the state of Mato Grosso were also found, on the border of Goiás State (Fig. 3D). In these locations it grows from 400 and 700 m a.s.l. *Heteropterys tocantinensis* is associated with clay soil (Fig. 6A), which is similar to the habitat of *H. walteri* and differs from *H. pannosa*, *H. rosmarinifolia*, and *H. veadeirensis*, which are associated with sandy soils. The species has been collected with buds and/or flowers in March and from September to January and with fruits from September to November. Although *H. tocantinensis* has a broad extant of occurrence, for now it should be considered Endangered (EN) [B2ii], because of its area of occupancy is less than 40 km$^2$. New expeditions in areas with clay soil in bordering states might reveal additional populations of *H. tocantinensis*.

### Paratypes

Brazil. Bahia: Formosa do Rio Preto, ESEC Serra Geral do Tocantins, ca. 15 km NE da Vila dos Prazeres, 10°42′58″S, 45°58′22″W, ca. 692 m, 3 October 2018 (fl), *M.F. Simon et al.* 3430 (CEN, RB); Estrada Formosa do Rio Preto para Mateiros, Fazenda Bom Jesus, 63 km de Formosa do Rio Preto, 10°37′18″S, 45°26′55″W, ca. 700 m, 2 March 2015 (bud, fl), *A.M. Amorim et al.* 9196 (CEPEC, RB); Mato Grosso: Mun. Barra do Garças, Parque Estadual da Serra Azul, 15°50′33″S, 52°16′49″W, 18 November 2008 (old fr), *E.S. Medeiros et al.* 517 (CEPEC, RB); Mun. Torixoréu, 15°59′S, 52°22′W, 24 October 1977 (fr), *J.S. Costa* 67 (RB); Tocantins: Mun. Dianópolis, 11°37′02″S, 46°24′53″W, ca. 628 m, 28 September 2003 (bud, fl), *T.B. Cavalcante et al.* 3250 (CEN); Ponto 404, 11°33′35″S, 46°28′48″W, ca. 670 m, 24 September 2003 (bud), *A.O. Scariot et al.* 672 (CEN); 11°36′48″S, 46°26′31″W, 27 September 2003 (fr), *A.O. Scariot et al.* 902 (CEN, CEPEC); Mun. Mateiros, Estação Ecológica Serra Geral do Tocantins, próximo a terra do Posseiro Manelão, 10°46′14″S, 46°43′10″W, ca. 461 m, 31 January 2015 (fl), *G.M. Antar et al.* 748 (SPF); Região do Jalapão, estrada Mumbuca a Boa Esperança, próximo ao posto Naturantins, 10°23′38″S, 46°36′46″W, 8 December 2005 (old fl), *G.H. Rua et al.* 680 (CEN); Jalapão, próximo ao Rio Pedro de Amolar, 9 September 1995 (old fl), *M. Alves* 1062 (CEPEC, HPN*); Mun. Pindorama do Tocantins, ca. 37.4 km da BR-010, 11°20′49.56″S, 47°36′7.9″W, 5 October 2007 (fl), *J. Paula-Souza et al.* 8932 (CTES*, SI*, SPF).

### Etymology

The specific epithet refers to the occurrence of the new species near the Tocantins River basin in central Brazil.

*Remarks*

For some vegetative characters, such as shape, consistency and size of the lamina, *Heteropterys tocantinensis* resembles *H. pannosa* and *H. walteri*. *Heteropterys tocantinensis* can be differentiated by the stems and leaves covered by a sericeous indumentum, which is early caducuous (*vs* stems and leaves glabrous in *H. veadeirensis* and densely and persistently hispid in *H. walteri*). Also, in *H. tocantinensis* the peduncle is strongly reduced, the anthers are pilose and the mericarp has a small dorsal wing, characteristics not observed in other species of this complex. Most records of *H. tocantinensis* are from Tocantins State but this species is not sympatric with *H. walteri*, which has an occurrence further south of that state (see Figs. 3D, 3F). Herbarium collections of *H. tocantinensis* were often misidentified as *H. byrsonimifolia* A. Juss., a very common species, which generally grows in highland savannas and on rock outcrops but has an arborescent habit and paniculate inflorescence.

*Heteropterys veadeirensis* **Amorim & Francener** *sp. nov.*
(Figs. 6C, 6D and 7)

*Type*

Brazil. Goiás: Mun. Alto Paraíso de Goiás, estrada Alto Paraíso a Colinas, ca. 35 km de Alto Paraíso, próximo a São Jorge, 14°10′S, 47°49′W, 2 August 2000 (bud, fl, fr), *R.C. Forzza et al.* 1671 (Holotype: SPF!; Isotypes: CEPEC!, RB!, UESC!).

*Diagnosis*

*Heteropterys veadeirensis* differs from the other species in the *H. pannosa* complex in its glabrous stems, petiole and lamina (*vs* densely hispid or sericeous to sparsely sericeous), very sparsely sericeous to glabrate peduncle and pedicel (*vs* densely hispid, sericeous or tomentose), and styles dorsally short-hooked at the apex (*vs* obtuse, truncate or slightly apiculate, except in *H. walteri*).

*Description*

Subshrub, 0.2–0.4 m tall, stems erect, cylindrical, ca. 1.5 mm diam., glabrous, lenticels not seen, unbranched or nearly so, all arising from a xylopodium. Leaves opposite; petiole 1–2.5 mm long, glabrous, eglandular; stipules minute protuberances, ca. 0.3 mm long, apparently absent from old leaves; lamina of larger leaves (2.4–)5.5–9.7 cm long, (2.5–)4.3–5.9 cm wide, glabrous, subcoriaceous to coriaceous, oblong to ovate, rarely elliptic, the base rounded or cordate, the apex obtuse, rounded or rarely acute, the margins entire, slightly revolute, sometimes with 1–2 impressed glands abaxially near the base and smaller glands irregularly scattered throughout lamina, the lateral veins and reticulum prominent on both surfaces. Inflorescence a pseudoraceme, axillary or terminal, erect, (5.2–)8.2–12.6(–17) cm long, glabrous or very sparsely sericeous on the distal part of the rachis, with irregular internodes between each pair of flowers, mostly comprising 6–12 flowers distributed throughout the rachis; bracts 3–3.5 mm long, 0.5–0.7 mm wide, linear-lanceolate, abaxially sericeous, margins entire, eglandular; peduncle (3–)6–9 mm long, very sparsely sericeous to glabrate; bracteoles apical, 1–1.5 mm long, 0.5–0.7 mm

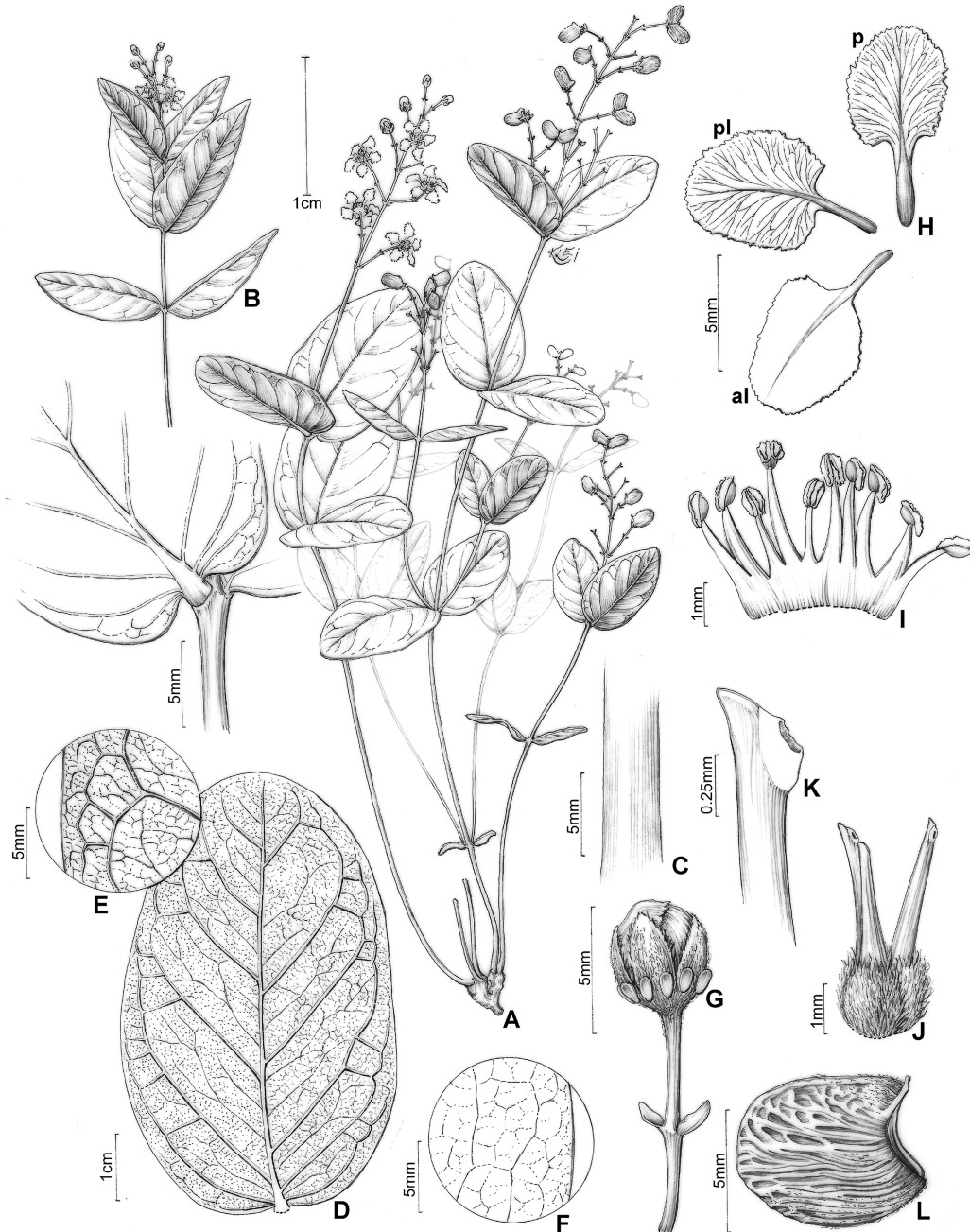

**Figure 7** **H. veadeirensis.** *Heteropterys veadeirensis.* (A) Habit with a xylopodium, showing enlargement of stem and leaf base. (B) Flowering branch from different individual. (C) Detail of stem. (D) Leaf in abaxial view. (E) Details of leaf margin in abaxial view. (F) Details of leaf margin in adaxial view. (G) Floral bud in lateral view. (H) Petals: posterior–p and posterior-lateral–pl in adaxial view; anterior-lateral–al in abaxial view. (I) Androecium in adaxial view, the stamen second from right opposite the posterior petal, the stamen fourth from left opposite the anterior sepal. (J) Gynoecium showing anterior style at center. (K) Detail of stigma. (L) Mericarp in lateral view. (A, C–L from Forzza 1671, B from Faria 1499, by Klei Sousa).

wide, ovate to lanceolate, abaxially sericeous, persistent, minutely glandular at margin; pedicel 5–6 mm long, 0.4–0.6 mm wide, uniformly slender, very sparsely sericeous to glabrate. Sepals 2.7–3 mm long, 1.5–2.5 mm wide, minutely oblong, acute at apex, revolute at anthesis, not appressed against filaments at anthesis, abaxially sparsely sericeous-ferrugineous, adaxially glabrous, the anterior sepal eglandular, the four lateral sepals biglandular, the glands 1–2 mm diam. Petals not exposed in the enlarging bud, vivid yellow, glabrous, membranaceous, not keeled, erose and eglandular at the margin, the posterior-lateral and anterior-lateral petals similar to each other, spreading, the claw 3–5 mm long, the limb 5.7–7 mm long, 2.8–5 mm wide; posterior petal spreading, the claw 4–6 mm long, the limb 5–5.5 mm long, 4.5–5.5 mm wide. Stamens glabrous; filaments slightly heteromorphic, longer opposite sepals than opposite petals, 3–3.5 mm long, 0.3–0.5 mm wide, all straight and slender, basally connate; anthers 0.7–1 mm long, slightly reflexed at anthesis, all alike; the connective proximally dark brown, distally yellow. Ovary 1.5–2 mm tall, minutely sericeous; styles 2.5–3 mm long, slightly equal, equaling or slightly exceeding the anthers, all erect and straight, glabrous, dorsally short-hooked at apex; stigmas lateral, all three facing the anterior sepal. Mericarp with ellipsoidal or rounded nut, 6–9 mm long, 5–6 mm wide, with several parallel longitudinal veins on each side, sericeous to glabrate; lateral wings or crests absent; dorsal wing absent or strongly reduced to an apical crest, a crest arising slightly at the style, ca. 1 mm wide; ventral areole 3–4 mm tall, ca. 2 mm wide, ovate.

### Distribution, phenology and conservation status

*Heteropterys veadeirensis* is restricted to northern Goiás State (Fig. 3E) and grows on rock outcrops (Fig. 6C) between 1.000 and 1.200 m a.s.l. This species has been collected with buds and/or flowers from July to November and with fruits from August to October. Almost all records were recorded from a protected area called Chapada dos Veadeiros National Park; the only exception is a collection made about 300 km to the south, in the municipality of Cristalina. Although these populations are protected and there is a considerable distance between the two localities, the species is assessed as Endangered (EN) according to *IUCN Standards & Petitions Subcommittee (2017)* criteria. The extent of occurrence is less than 4,000 km$^2$ and area of occupancy less than 28 km$^2$. We still do not know if the specimens from Alto Paraíso de Goiás belong to only one population, but it is likely that there are fewer than five populations [B1 + B2a].

### Paratypes

Brazil. Goiás: Mun. Alto Paraíso de Goiás, 14°04′14″S, 47°54′54″W, 25 September 1995 (fr), *M.L. Fonseca & F.C.A. Oliveira 547* (RB, SPF); Chapada dos Veadeiros, Estrada entre Alto Paraíso e São Jorge, 14°08′08″S, 47°43′21″W, 15 October 2010 (bud, fl), *A. Francener et al. 1009* (CEPEC, UB, UFG, UFMT*); Estrada para o Vale da Lua, 14°10′25″S, 47°47′05″W, 15 October 2010 (bud, fl, fr), *A. Francener et al. 1021* (CEPEC, CGMS*, UFG); próximo a sede do Parque Nacional da Chapada dos Veadeiros, 14°09′32″ S, 47°47′41″W, 16 October 2010 (bud, fl), *A. Francener et al. 1025* (CEPEC, UFG); Parque Nacional da Chapada dos Veadeiros, 14°10′29.3″S, 47°49′25.7″W, 3 October 2007

(fr), *J. Paula-Souza et al.* 8817 (SPF); 3 October 2007 (fl), *J. Paula-Souza et al.* 8818 (SPF); Rodovia GO-239 em direção a São Jorge, 32.5 km do entroncamento com a GO-118, 14°09′47.2″S, 47°46′48.7″W, 4 October 2007 (fl), *J. Paula-Souza et al.* 8872 (SPF); 4 October 2007 (fr), *J. Paula-Souza et al.* 8882 (SPF); Vale da Lua, trilha para o Rio São Miguel, 14°11′19.95″S, 47°47′27.2″W, 21 October 1996 (fr), *R. Marquete et al.* 2737 (RB); Mun. Cristalina, Lages, ca. 12 km ao sul de Cristalina, 16°52′20″S, 47°37′02″W, ca. 966 m, 30 July 2011 (fl), *J.E.Q. Faria et al.* 1499 (CEN, HUEG*, UB); Mun. São João da Aliança, Parque Nacional da Chapada dos Veadeiros, 29 November 1988 (old fl), *T.B. Cavalcante et al.* 26 (SPF).

### Etymology
The specific epithet refers to *Chapada dos Veadeiros*, a wide formation of mountains where this species is found. This region probably has the highest diversity of Malpighiaceae on earth, including several endemic species in the group.

### Remarks
*Heteropterys veadeirensis* is distinguished from all other species in the *H. pannosa* complex by its glabrous vegetative morphology. All the other species have simple and basifixed or Y-V-T-shaped trichomes on the stems and leaves. The shape and size of *H. veadeirensis* leaves resemble the old leaves of *H. tocantinensis* but most collections of that species have a distinctive indumentum. *Heteropterys veadeirensis* is also distinguished by its lamina base that sometimes has 1–2 impressed glands abaxially and smaller glands irregularly scattered throughout lamina. One isolated population know only from one specimen (*Faria* 1499) from an area in the southern part of the Brazilian Federal District differs from all other specimens by having an elliptical lamina shape (Fig. 7B). More representative collections from this area are needed to confirm this difference.

### *Heteropterys walteri* Amorim & Francener *sp. nov.*
(Fig. 8)

### Type
Brazil. Goiás: Município Niquelândia, Estrada Niquelândia a Uruaçu, ca. 50 km Uruaçu, 14°19′42″S, 48°06′29″W, 15 July 2000 (bud, fl, fr), *V.C. Souza et al.* 23900 (Holotype: ESA!; Isotypes: CEN!, CEPEC!, RB!, SP!, UESC!).

### Diagnosis
*Heteropterys walteri* differs from the remaining species in the *H. pannosa* complex in its stems, petiole, leaf midrib, inflorescence rachis, peduncle and pedicel densely and persistently hispid (*vs* densely or sparsely sericeous to glabrous) and abaxial and adaxial lamina surfaces densely tomentose (*vs* abaxial and adaxial lamina surfaces densely or sparsely sericeous to glabrous).

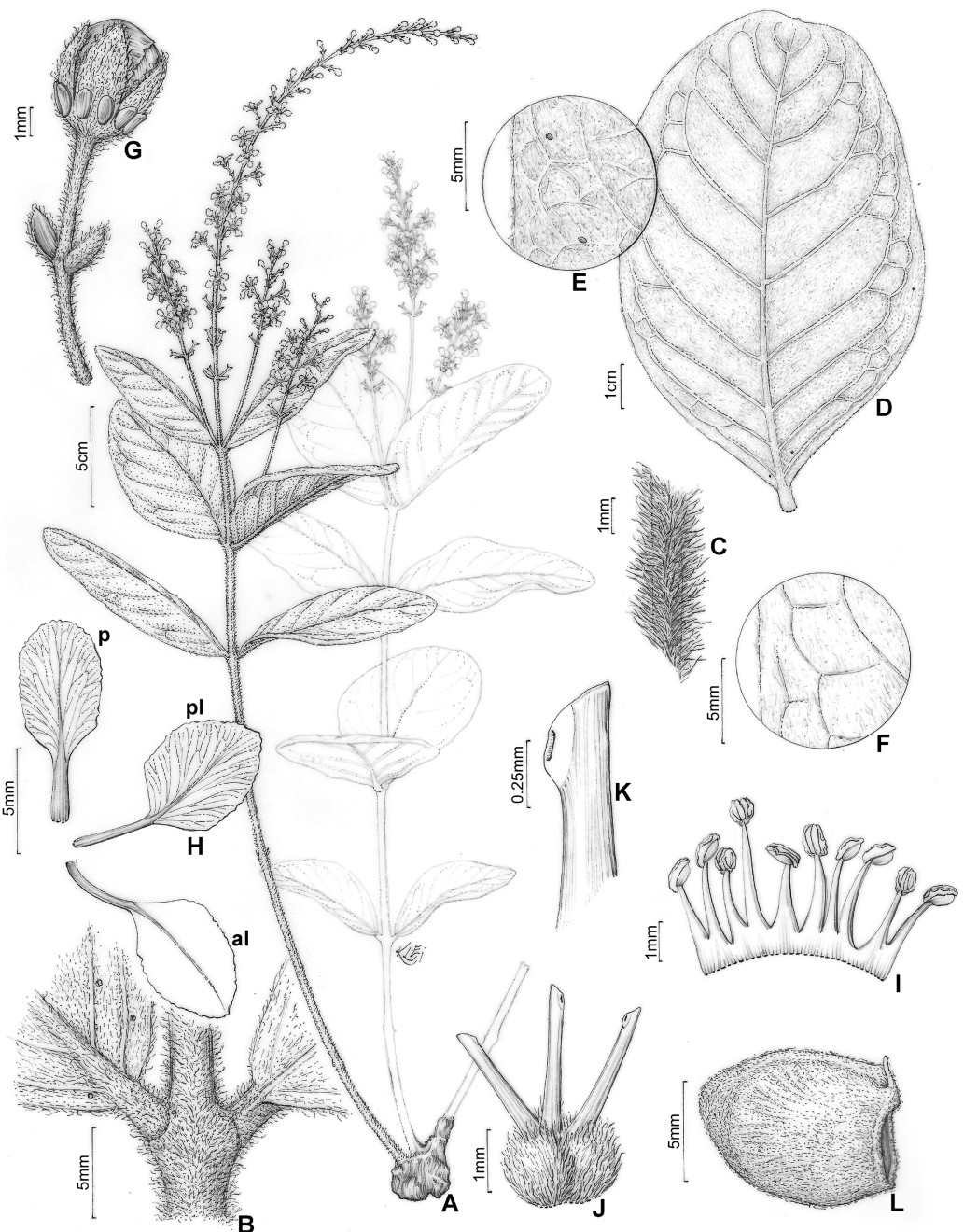

**Figure 8** *H. walteri.* *Heteropterys walteri*. (A) Habit with a xylopodium. (B) Detail of stem and leaf base. (C) Detail of stem showing the indumentum. (D) Leaf in abaxial view. (E) Details of young leaf margin showing the glands, abaxial view. (F) Details of old leaf margin in adaxial view. (G) Floral bud in lateral view. (H) Petals: posterior–p and posterior-lateral–pl in adaxial view; anterior-lateral–al in abaxial view. (I) Androecium in adaxial view, the stamen second from right opposite the posterior petal, the stamen fourth from left opposite the anterior sepal. (J) Gynoecium showing anterior style at center. (K) Detail of stigma. (L) Mericarp in lateral view. (A–K from Souza 23900, L from G. Pereira-Silva 5389, by Klei Sousa).

*Description*

Subshrub, 0.3–0.5 m tall, stems erect, cylindrical, ca. 4 mm diam., densely hispid to glabrate with age, lenticels not seen, unbranched or nearly so, all arising from a xylopodium. Leaves opposite or rarely 3-whorled on the same stem; petiole 3–5 mm long, densely hispid, eglandular; stipules minute, ca. 0.5 mm long, hidden by indumentum; lamina of larger leaves (6.7–)8–11.8(–14) cm long, 3.2–7.2 cm wide, abaxially and adaxially densely tomentose, glabrescent with age, the midrib and primary veins on both surfaces densely hispid, coriaceous, oblong or slightly obovate, rarely elliptical, the base cuneate or rarely obtuse, the apex rounded and often mucronate, the margins entire, sometimes with two large glands abaxially at base and usually with 7–13 smaller impressed glands in an inframarginal row on each side of the lamina or rarely glands absent, the glands ca. 0.5 mm diam., the lateral veins and reticulum prominent on abaxial surface. Inflorescence a pseudoraceme, mostly elongate, axillary or terminal, erect, 8–16(–19.5) cm long, densely hispid, with measurable and irregular internodes between each groups of 2–3 flowers, mostly comprising 14–21 flowers distributed throughout the rachis; bracts 2–3 mm long, ca. 1 mm wide, abaxially densely hispid to tomentose, linear-lanceolate, persistent, margins entire, eglandular or with 1–2 glands near the base, the glands ca. 0.5 mm diam., generally hidden by indumentum; peduncle 5–8 mm long, ca. 1 mm wide, densely hispid; bracteoles apical, 1.7–2 mm long, ca. 0.5 mm wide, abaxially densely hispid to tomentose, linear-lanceolate, persistent, eglandular or with 1–2 glands near the base, the glands ca. 0.3 mm diam., hidden by indumentum; pedicel 5.5–7 mm long, ca. 1 mm wide, uniformly slender, densely hispid. Sepals 3.5–4 mm long, 1.5–2 mm wide, narrowly ovate, acute at apex, revolute at anthesis, not appressed against filaments at anthesis, abaxially densely tomentose, adaxially glabrous, the anterior sepal eglandular, the four lateral sepals biglandular, the glands 1–1.5 mm diam. Petals not exposed in the enlarging bud, vivid yellow, glabrous, membranaceous, not keeled, erose and eglandular at the margin, the posterior-lateral and anterior-lateral petals similar to each other, spreading, the claw 3–4 mm long, the limb 3.5–4.5 mm long, 3.5–4.5 mm wide; posterior petal spreading, the claw 3.5–4 mm long, the limb 5–5.3 mm long, 4–4.4 mm wide. Stamens with filaments heteromorphic, longer opposite sepals than opposite petals, glabrous, 2.5–3 mm long, 0.3–0.5 mm wide, all straight and slender, basally connate; anthers 0.8–1 mm long, glabrous, slightly reflexed at anthesis, all alike; the connective proximally dark brown, distally yellow. Ovary 1.3–1.5 mm tall, densely sericeous-ferrugineous; styles 1.7–2 mm long, slightly unequal, equaling or slightly exceeding the anthers, the anterior style erect and straight, the two posterior styles divergent, glabrous, dorsally short-hooked at apex; stigmas lateral, all three facing the center of the flower. Mericarp with rounded nut, 9–12 mm long, 7–10 mm wide, with several parallel longitudinal veins on each side, densely sericeous-ferrugineous, the trichomes persistent; lateral wings or crests absent; dorsal wing absent or strongly reduced to an apical crest, a crest arising slightly at the style, 1–3 mm wide, widest near distal side of nut; ventral areole 3–5 mm tall, 3–5 mm wide, ovate.

*Distribution and conservation status*

*Heteropterys walteri* occurs on the border of the states of Goiás and Tocantins (Fig. 3F) and is generally associated with clay soil between 280 and 940 m a.s.l. This species has been collected with buds and/or flowers in March and from July to December and with fruits in March and from July to November. Most specimens were collected along highways and roads, although many collections were recently obtained in the Serra do Tombador Natural Reserve Protection Area in Goiás State. Based on an area of occupancy less than 96 km$^2$, *H. walteri* can be considered Endangered (EN). The list of paratypes is long but most of these collections are from outside protected areas and close to urban areas and roads, which affect the habitat quality [B2bii, iii].

*Paratypes*

Brazil. Goiás: Mun. Campinaçu, Estrada Niquelândia a Campinaçu, 14°03′S, 48°30′W, ca. 420 m, 6 October 1995 (bud, fl), *T. B. Cavalcante et al.* 1812 (CEN); Região da Fazenda Praia Grande, ca. 6 km após o córrego Praia Grande, 13°59′S, 48°23′W, ca. 430 m, 6 October 1995 (bud, fl), *B.M.T. Walter et al.* 2680 (CEN); Mun. Cavalcante, Estrada Balsa Porto dos Paulistas, Rio Tocantins, Buracão e Curral de Pedra, a ca. 5.8 km do rio, 13°27′43″S, 48°07′16″W, ca. 410 m, 9 November 2000 (old fl), *G. Pereira-Silva et al.* 4379 (CEN, CEPEC); Estrada Canteiro da Obra do Rio São Félix, Km 12, 13°22′32″S, 48°03′49″W, ca. 430 m, 19 September 2001 (fr), *G. Pereira-Silva et al.* 5389 (CEN, CEPEC); Estrada de acesso ao Rio Traíras, ca. 9 km da cidade, 12°20′07″S, 48°08′33″W, ca. 350 m, 27 November 2007 (fr), *G. Pereira-Silva* 12375 (CEN); Estrada entre Cavalcante e Minaçú, 77 km de Cavalcante, 13°38′06″S, 47°48′07″W, ca. 860 m, 24 July 2014 (bud, fl), *M.F. Simon* & *L.M. Borges* 2473 (CEN); RPPN Serra do Tombador, ca. 12 km no sentido a Cavalcante, 13°38′02″S, 47°44′46″W, ca. 908 m, 11 November 2014 (fr), *M. Mendoza et al.* 4385a (CEN); Reserva Natural da Serra do Tombador, Campina, 13°42′S, 47°47′W, 21 August 2017 (bud, fl), *H.L. Zirondi* 39 (CEN, HRCB*); Reserva Natural da Serra do Tombador, Estrada GO 241, Cavalcante a Minaçu, 3.5 km após a sede da reserva, 13°40′49″S, 47°49′12″W, 5 March 2017 (fr), *C.O. Andrino et al.* 406 (CEN); Mun. Colinas do Sul, Estrada pelo dique 2 na direção do Rio Tocantins, próximo a Serra da Mesa/Colinas, 13°53″S, 48°19′W, ca. 410 m, 20 October 1996 (fr), *B.M.T. Walter et al.* 3487 (CEN); Mun. Minaçu, Serra da Mesa, a 7 km do canteiro de obras, 13°34′S, 48°10′W, ca. 840 m, 11 October 1991 (old fl), *T.B. Cavalcante et al.* 1027 (CEN); Mun. Niquelândia, Embarcadouro da CODEMIM, após o portão principal, 14°08′54″S, 48°19′34″W, 14 December 1998 (old fl), *A.A. Santos et al.* 371 (CEN, CEPEC); Estrada Niquelância a Indaianópolis, ca. 10 km sw Indaianópolis, 14°14′29.9″S, 48°32′36.1″W, 11 September 1998 (old fl), *V.C. Souza et al.* 21528 (ESA); Estrada Niquelândia a Colinas, 14°21′S, 48°06′W, 17 September 1998 (bud, fl), *E.L. Jacques et al.* 796 (CEN, SP); Estrada Niquelândia a Uruaçu, ca. 25 km de Uruaçu, 14°19′42″S, 48°06′29″W, 12 September 1998 (bud, fl), *V.C. Souza et al.* 21569 (CEPEC, ESA, RB); Estrada de chão vicinal à Rodovia GO-132, 14°19′44″S, 48°06′33″W, ca. 557 m, 27 November 2014 (fr), *J.A. Oliveira et al.* 513 (CEN, CEPEC, RB); Mun. São João da Aliança, Serra Geral do Paranã, ca. 3 km de São João, 14°33′59″S, 47°22′38″W, ca. 850 m, 16 March 1971 (bud, fl), *H.S. Irwin et al.* 31940

(NY, UB). Tocantins, Mun. Paranã, Canteiro de Obras do UHE São Salvador, 12°48′18″S, 48°13′59″W, ca. 260 m, 19 October 2006 (bud, fl, fr), *G. Pereira-Silva et al.* 10898 (CEN); Estrada de acesso ao vilarejo Rozario, ca. 3 km após a entrada principal da obra, 12°47′42″S, 48°11′58″W, 24 March 2007 (old fl); *G. Pereira-Silva et al.* 11462 (CEN); 12°49′33″S, 48°13′14″W, 27 September 2007 (bud, fl); *G. Pereira-Silva et al.* 12111 (CEN, CEPEC).

### Etymology

The epithet honors Dr. Bruno Machado Teles Walter, a researcher at the Embrapa Recursos Genéticos e Biotecnologia (CENARGEN) and Curator at the CEN herbarium. Besides collecting some of the paratypes, he has conducted several important studies about community structure in the Cerrado domain.

### Remarks

*Heteropterys walteri* is notable for its stems, petiole, leaf midrib, inflorescence rachis, peduncle and pedicel covered by a densely hispid indumentum, which is mostly specially formed by simple and basifixed trichomes. In some collections, especially those with old flowers and fruits, the lamina indumentum on both surfaces is glabrescent, which often makes it difficult to distinguish this species from some relatives. Within this difficult species complex, *H. walteri* is also distinguished by its long inflorescence rachis (reaching up to 19.5 cm) with more than 20 flowers in some specimens. This species resembles *H. tocantinensis* in leaf shape (for more detail, see comments above). Herbarium collections of *H. walteri* are often misidentified as *H. duarteana* A. Juss., a species that is also present in the Cerrado domain but has a shrubby habit without a xylopodium, a large paniculate inflorescence and mericarps with a well-developed dorsal wing.

### Funding

André M. Amorim received a Research Productivity Fellowship from the Conselho Nacional de Desenvolvimento Científico e Tecnológico-CNPq (#312404/2018-2) and laboratory work was supported by CNPq (Edital Universal, #436283/2018-2). The funders had no role in study design, data collection and analysis, decision to publish, or preparation of the manuscript.

### Grant Disclosures

The following grant information was disclosed by the authors:
Conselho Nacional de Desenvolvimento Científico e Tecnológico-CNPq: #312404/2018-2.
CNPq: #436283/2018-2.

### Competing Interests

The authors declare that they have no competing interests.

## Author Contributions

- André M. Amorim conceived and designed the experiments, performed the experiments, analyzed the data, prepared figures and/or tables, authored or reviewed drafts of the paper, paid financial expenses for the illustrations and visits to the collections, and approved the final draft.
- Lucas C. Marinho performed the experiments, analyzed the data, prepared figures and/or tables, authored or reviewed drafts of the paper, and approved the final draft.
- Augusto Francener performed the experiments, analyzed the data, authored or reviewed drafts of the paper, and approved the final draft.

## Data Availability

As it is a plant systematics manuscript, all material used to prepare the work is available in the results.

## New Species Registration

The following information was supplied regarding the registration of a newly described species:

Heteropterys tocantinensis Amorim & Francener: 77254271-1

Heteropterys veadeirensis Amorim & Francener: 77254272-1

Heteropterys walteri Amorim & Francener: 77254273-1

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
