# Peer review of "Deciphering the Heteropterys pannosa species complex (Malpighiaceae)"

_PeerJ, doi:10.7717/peerj.12937_

## Round 0.1 · original submission · Minor Revisions

Dear authors

As you will see, the two reviewers ask only minor revisions for your manuscript. I subscribe to that view. The requested changes should be easily implemented.

Best wishes
Mike Thiv

·

Basic reporting

no comment

Experimental design

no comment

Validity of the findings

no comment

Additional comments

The Heteropterys pannosa complex has perplexed botanists for generations, and the resolution presented here is a valuable contribution to the understanding of the Brazilian flora. It was a pleasure to read this exceptionally well-prepared manuscript. The illustrations are of fine quality (but see notes below). I have made a few suggestions for the text with the “comments” tool on the pdf. Below are some detailed notes.

Materials and Methods. The authors cite eleven herbaria, but many more are listed in the sections of specimens examined. It is customary to list under “Acknowledgments” all the herbaria that contributed to the study. Perhaps a note of explanation would clarify why only the eleven herbaria are listed.

H. tocantinensis. Throughout the treatment of this novelty the authors emphasize that the pedicel is greatly reduced and only about half as long as the pedicel. Yet, in Fig. 5 both in the habit sketches (A, B) and the detail (F) the peduncle and pedicel are roughly equally long.

Illustrations and captions. The drawings of the androecia confused me. They appear not to be bilaterally symmetrical; is that correct? It would be helpful to note in the caption the position of the stamens relative to the perianth, at least by noting which stamen opposes the anterior sepal (the longest one?).— Throughout the captions the word “nut” should be changed to “mericarp.”

Reviewer 2 ·

Basic reporting

no comment

Experimental design

no comment

Validity of the findings

no comment

Additional comments

1- Organize the description of the genus by parts: 1) morphology, 2) systematic and 3) distribution (line 41, move the habit to morphological description).
2. Line 51, change word work.
3. Line 52, different habitats? Change word, too ambiguous, preferably delete.
4. I suggest that in the sentences that go from line 59 to 69, the examples are reduced and only the consulted investigations remain.
5. Appointment for the sentences of line 73 and 75?
6.Capitalize the letters of the figures that appear in the text. They appear in capital letters in the figure captions, as in the image.
7. Line 114 and 120 change work for studie.
8. Line 121 add (“).
9. Line 147 Goyas or Goiás?
10. I suggest a brief description of the genus Heteropterys
11. Line 158 margin in singular without the the
12. The description of the indumentum of lamina is missing, only that of the margin appears. It's confusing.
13. Description of bract indumentum is lacking. In H. pannosa.
14. In materials and methods it is not mentioned that the HDJF herbarium was reviewed, in line 199 it is cited. Like NY, IBGE, HEPH and MBM.
15. line 240 put first “after.
16. Is the elevation and biome known where H. pannosa is distributed? Add.
17. In all descriptions it appears as stems erect and cylindrical, I suggest removing it.
18. In line 279 put margin in singular. The description of the outfit seems confusing, you only have outfit in the margin?
18. Why is Francener listed as the author of all new species and is not in the article? I suggest adding it.
19. In H. tocatins, lines 367 and 368, the arrangement of the description of the characters is different from the previous ones, homogenize: size, shape, margins, eglandular or not, persistent or not. Indument is lacking for bract descriptions of H. rosmarinifolia. Bract indumentum is also lacking.
20. add the indumentum of the stipulates if they are present.
21. In the descriptions of all species, accommodate the description of characters for stamens and styles, size, shape, clothing. Homogenize.
22. All the collectors have their abbreviated names, homogenize in line 404.
23. Homogenize with the previous descriptions the characters of the bracts and bracts in H. walteri.
24. Figure 1, manifolds in italics.
25. figure 3, add space between H. pannosa and complex
26. Figure 3 I suggest adding caption for each image, it would be much better to interpret.
27. Figure 3, remove the citation from the text, it already appears in materials and methods.
28. Figure 4, manifolds in italics.
29. Homogenize the bibliographic references, in some citations the volume is added and in others not.

---

## Round 0.2 · accepted · Accept

Dear authors

I went through your revised text. Except for some minor points it is acceptable. I attach a pdf with changes tracked in BLUE. These should still be considered.

Several specimens were collected in nature reserves. Do you have any collection permits? In case you should refer to those in MM.

Nice study, interesting plants, good drawings.

Best wishes
Mike Thiv